# Breaking the Fear Barrier: Aberrant Activity of Fear Networks as a Prognostic Biomarker in Patients with Panic Disorder Normalized by Pharmacotherapy

**DOI:** 10.3390/biomedicines11092420

**Published:** 2023-08-29

**Authors:** Haohao Yan, Yiding Han, Xiaoxiao Shan, Huabing Li, Feng Liu, Ping Li, Jingping Zhao, Wenbin Guo

**Affiliations:** 1Department of Psychiatry, National Clinical Research Center for Mental Disorders, National Center for Mental Disorders, The Second Xiangya Hospital of Central South University, Changsha 410011, China; yanhaohao1995@gmail.com (H.Y.); yidinghan1997@gmail.com (Y.H.); shanxiaoxiao22@126.com (X.S.); zhaojingpingcsu@163.com (J.Z.); 2Department of Radiology, The Second Xiangya Hospital of Central South University, Changsha 410011, China; huabingli3350@163.com; 3Department of Radiology, Tianjin Medical University General Hospital, Tianjin 300052, China; fengliu@tmu.edu.cn; 4Department of Psychiatry, Qiqihar Medical University, Qiqihar 161006, China; lipingchxyy@163.com

**Keywords:** panic disorder, magnetic resonance imaging, brain, biomarker, fear, anxiety, longitudinal studies, follow-up studies, machine learning, paroxetine

## Abstract

Panic disorder (PD) is a prevalent type of anxiety disorder. Previous studies have reported abnormal brain activity in the fear network of patients with PD. Nonetheless, it remains uncertain whether pharmacotherapy can effectively normalize these abnormalities. This longitudinal resting-state functional magnetic resonance imaging study aimed to investigate the spontaneous neural activity in patients with PD and its changes after pharmacotherapy, with a focus on determining whether it could predict treatment response. The study included 54 drug-naive patients with PD and 54 healthy controls (HCs). Spontaneous neural activity was measured using regional homogeneity (ReHo). Additionally, support vector regression (SVR) was employed to predict treatment response from ReHo. At baseline, PD patients had aberrant ReHo in the fear network compared to HCs. After 4 weeks of paroxetine treatment (20 mg/day), a significant increase in ReHo was observed in the left fusiform gyrus, which had shown reduced ReHo before treatment. The SVR analysis showed significantly positive correlations (*p* < 0.0001) between the predicted and actual reduction rates of the severity of anxiety and depressive symptoms. Here, we show patients with PD had abnormal spontaneous neural activities in the fear networks. Furthermore, these abnormal spontaneous neural activities can be partially normalized by pharmacotherapy and serve as candidate predictors of treatment response. Gaining insight into the trajectories of brain activity normalization following treatment holds the potential to provide vital insights for managing PD.

## 1. Introduction

Panic disorder (PD) is a prevalent type of anxiety disorder that is characterized by sudden and recurrent panic attacks [1]. Approximately 13.2% of individuals experience panic attacks at some point in their lifetime, with 12.8% of those meeting the criteria for PD as outlined in the Diagnostic and Statistical Manual of Mental Disorders, Fifth Edition (DSM-5) [2]. PD can have a significant impact on an individual’s quality of life and daily functioning, as well as impose significant individual and social burdens [3]. Despite ongoing research efforts, the underlying causes of PD are still not fully understood.

The etiopathogenesis of panic disorder involves a complex interplay of genetic, neurobiological, psychological, and environmental factors [4]. Although the precise cause remains elusive, there are several prominent factors that have been pinpointed. These factors encompass genetics, imbalances in neurotransmitters, anomalies in brain circuits, as well as the impact of stress and trauma [5,6,7,8,9]. There are various proposed hypotheses on the pathogenesis of PD, including cognitive, behavioral, and biological perspectives [10,11,12]. One of the most influential models is the fear network model [13,14], which originally linked panic attacks, anticipatory anxiety, and phobic avoidance to three brain regions: the brainstem, limbic system, and prefrontal cortex [14]. The rationale behind this linkage is as follows: (1) the occurrence of spontaneous panic attacks, manifesting with a range of autonomic symptoms, strongly suggests a central role played by the brainstem; (2) the realm of anticipatory anxiety is intricately tied to the limbic regions. Interventions such as benzodiazepines, mindfulness breathing techniques, and relaxation exercises function by attenuating limbic system activity, thereby alleviating anticipatory anxiety [15]; (3) the domain of phobic avoidance intricately involves conscious cognitive learning, overseen by the prefrontal cortex [16,17]. However, the revised hypothesis proposes a fear network centered on the amygdala, which involves various brain regions, including the thalamus, hypothalamus, hippocampus, frontal cortex, and sensory cortex, as well as the locus coeruleus, periaqueductal gray region, and other brainstem sites [13]. Elevated amygdala activity resulting from a breakdown in the coordination and transmission of sensory information from both “upstream” (cortical) and “downstream” (brainstem) sources leads to behavioral, autonomic, and neuroendocrine activation [13]. Concurrently, inhibitory influences from the frontal cortex are diminished, culminating in the emergence of panic attacks [13]. Dresler, et al. [18] have proposed a modified model of the fear network, building on the hypothesis of Gorman, Kent, Sullivan and Coplan [13]. In this revised model, more extensive brain areas, including the insula and anterior cingulate, are involved, and the hyperactive amygdala is considered a state characteristic, rather than a trait characteristic of PD [18]. Furthermore, Lai [19] recently introduced an advanced fear network model that includes dysregulation of the fronto-limbic-insula and temporal–occipital–parietal areas, particularly the sensory regions.

Abnormalities in the fear network have been identified using various neuroimaging methods, including functional near-infrared spectroscopy (fNIRS) [20], electroencephalography (EEG) [21], positron emission tomography (PET) [22,23], structural magnetic resonance imaging (MRI) [24], task-based fMRI [25], and resting-state fMRI [26,27]. Resting-state fMRI holds the capability to evaluate inherent brain connectivity and functional networks without necessitating specific participant tasks. This technique remains non-invasive, mitigating participant discomfort and permitting repeated measurements [27]. Additionally, resting-state fMRI boasts broad coverage of the entire brain, enabling the exploration of both global and localized network connectivity patterns across diverse brain regions. Unlike fNIRS and EEG, which possess comparatively lower spatial resolution, resting-state fMRI offers a heightened spatial resolution, thereby simplifying the identification of distinct brain regions engaged in connectivity [28]. Consequently, we chose resting-state fMRI to investigate the underlying pathogenesis of PD and to provide neuroscience evidence for the advanced fear network model. In this study, we employed regional homogeneity (ReHo) analysis, a widely used and highly reliable approach for analyzing resting-state fMRI data. ReHo analysis is a data-driven method that measures local functional connectivity (FC) throughout the brain, without requiring any prior hypothesis.

Furthermore, prior studies have reported treatment-related brain changes in patients with PD [25,26,29,30,31]. Prospective studies offer insights not only into the therapeutic mechanisms of interventions for PD but also into the understanding of the pathogenesis of PD. However, these brain changes have only been observed in patients undergoing psychodynamic treatment [25,30,31], studies using task-based fMRI [25,30,31], and studies using resting-state fMRI design and pharmacotherapy [26,29], but all with small sample sizes [26,29]. Therefore, a longitudinal resting-state fMRI study with a relatively large sample size is necessary to investigate the pathogenesis of PD and to discern the trajectory of brain function changes in patients with PD after pharmacotherapy. Consequently, the present resting-state fMRI study employed a longitudinal design. Another pivotal decision pertains to the choice of intervention. Selective serotonin reuptake inhibitors (SSRIs) are the standard first-line pharmacological treatments for PD [26]. While SSRIs may intensify anxiety and panic during the early stages of treatment and have a delayed onset of action lasting several weeks, they are less likely to cause dependence and tolerance compared to benzodiazepines [27]. Therefore, in the current study, paroxetine, an SSRI commonly prescribed to patients with PD, was selected as the treatment option.

In addition to tracking the trajectory of brain function changes after treatment, a prospective study using resting-state fMRI could explore the feasibility of creating a predictive model based on spontaneous neural activity to predict the treatment response of patients with PD to pharmacotherapy. Accurately predicting treatment response could optimize the utilization of medical resources, minimize exposure to ineffective interventions, and enhance treatment compliance. Brain structural and functional features, such as blood oxygenation level-dependent (BOLD) signal [32,33] and gray matter volume (GMV) [33], as well as clinical features [34,35,36,37,38,39,40] like anxiety sensitivity, anxiety level, age of onset, agoraphobic severity, and childhood maltreatment, have shown potential in predicting the treatment response of patients with PD to pharmacotherapy or cognitive behavioral therapy (CBT). However, to the best of our knowledge, no study has investigated the use of resting-state brain activity patterns to predict the treatment response of PD patients to pharmacotherapy. Hence, in the present study, we employed support vector regression (SVR) analysis with abnormal ReHo values as the features to construct a predictive model for the treatment response of patients with PD to paroxetine. SVR, a classic machine learning method, could yield good performance in solving non-linear regression tasks (such as prediction of treatment response) by projecting the original feature (such as ReHo values) into kernel space [41]. Additionally, SVR has excellent generalizability [42].

The longitudinal resting-state fMRI study involved the enrollment of a sizable participant pool, encompassing individuals diagnosed with PD as well as healthy controls (HCs). The study design entailed the collection of both clinical data and fMRI data prior to the initiation of treatment. Subsequently, patients with PD underwent a 4-week treatment with paroxetine. Following the treatment period, fMRI scans and clinical symptom assessments would be administered to the patient group. The central objective of this longitudinal investigation was to assess the intrinsic neural activity patterns in individuals with PD and to discern their evolution subsequent to pharmacotherapy. Moreover, the study aimed to explore the potential of these neural activity patterns in predicting treatment response. Furthermore, the research endeavored to examine the correlation between ReHo and the baseline clinical symptoms of patients with PD and the correlation between the alterations in ReHo and the changes in clinical symptoms after treatment. The study hypothesized that (1) abnormal ReHo in the advanced fear network, including the fronto-limbic-insula and temporal–occipital–parietal areas, would be present in patients with PD compared to HCs; (2) some of the aberrant ReHo would be normalized after paroxetine treatment; (3) abnormal ReHo values would be associated with pretreatment clinical symptoms, and changes in ReHo values would correlate with changes in clinical symptoms after treatment; and (4) abnormal ReHo could predict early treatment effects of paroxetine in patients with PD.

## 2. Materials and Methods

### 2.1. Participants

The study was approved by the Ethics Committee of Second Xiangya Hospital, Central South University of China. The approval was granted under the registration number 2018025. Furthermore, the clinical trial was registered on ClinicalTrials.gov under the registration number NCT03894085. The trial’s registration information can be found at the following URL: https://clinicaltrials.gov/ct2/show/NCT03894085 (accessed on 20 August 2023).

Understanding the necessary number of subjects for a replicable or dependable analysis is crucial, preventing minor fluctuations in the subject pool from unduly influencing analysis outcomes. Earlier fMRI investigations have indicated that including approximately 30 subjects is advisable for fMRI group-level studies [43,44,45], contributing to the enhancement of result reproducibility as the subject count increases [46]. Considering our study’s longitudinal nature, the potential for participant attrition, and the need to exclude data-contributing participants from statistical analysis, we have chosen to engage a minimum of 50 participants per group. Additionally, prior to their participation, all individuals involved in the study provided written informed consent. Between January 2019 to August 2022, we recruited 57 drug-naive patients with PD from the outpatient department of the Department of Psychiatry of the Second Xiangya Hospital, as well as 56 HCs from the local community through advertising. PD diagnosis was ascertained by two psychiatrists using the Structured Clinical Interview for DSM-5 [1], while the non-patient edition of the Structured Clinical Interview for DSM-5 Axis I Disorders was used to screen the HCs [1]. HCs had no history of past or current psychiatric disorders, and there was no reported family history of any psychiatric disorders. The criteria for recruiting patients and HCs were as follows: (1) no prior exposure to psychiatric medication; (2) no contraindications for MRI scans; (3) no history of brain trauma or substance abuse disorder; (4) absence of other neurological or significant physical illnesses, including PD patients without accompanying psychiatric comorbidities; and (5) education level of at least 6 years and age between 18 and 60 years. We excluded patients who took benzodiazepines or other synergists into the paroxetine treatment. A detailed flow chart illustrating the participant selection process is provided in Figure 1.

### 2.2. Procedure

At the baseline (0 weeks) visit, we collected demographic information from both patients with PD and HCs and performed fMRI scans on them using a 3.0 T Philips scanner (Philip Medical Systems, Best, The Netherlands) located at the Second Xiangya Hospital. Data were obtained using the Achieva platform and an 8-element head coil array was utilized. Following the baseline scans, patients with PD received 4 weeks of treatment with paroxetine (20 mg/day) and underwent a second scan after the treatment period. We assessed the clinical symptoms of patients with PD at baseline and after treatment.

### 2.3. Measures

Prior studies have indicated that individuals with PD exhibit varying degrees of depressive and anxiety symptoms, alongside prevalent deficits in social and cognitive capabilities [47,48,49,50]. These patients tend to adopt diverse coping mechanisms for managing daily life events and often display distinctive personality traits [51,52]. Moreover, recent investigations have shown significant associations between emotional symptoms, social and cognitive impairments, personality traits, and corresponding patterns of brain activity in PD patients [48,49,50]. To assess anxiety and depressive symptoms in patients with PD, we used the Chinese version of the Hamilton Anxiety Rating Scale (HAMA, Cronbach’s alpha = 0.86) and the 17-item Hamilton Depression Rating Scale (HAMD, Cronbach’s alpha = 0.71), respectively [53,54]. Higher scores on these scales indicate greater severity of anxiety and depressive symptoms. We evaluated impairments in social and cognitive functions using the Social Disability Screening Schedule (SDSS, Cronbach’s alpha = 0.85) and the Brief Cognitive Assessment Tool for Schizophrenia (B-CATS, Cronbach’s alpha = 0.73), respectively [55,56]. The SDSS is a clinician-administered 10-item scale, with higher scores indicating greater impairment in social function. The B-CATS consists of four subscales—Digit Symbol Substitution Test (DSST), Trail Making Test-A (TMT-A), Trail Making Test-B (TMT-B), and Category Fluency (CF)—and lower DSST and CF scores, as well as higher TMT-A and TMT-B scores, indicate worse cognitive performance [57]. We used the Simplified Coping Style Questionnaire (CSQ, Cronbach’s alpha = 0.90) to assess different ways of coping with daily life events, which consists of two subscales: active coping (the first 12 items) and passive coping (the last 8 items) [58]. Finally, we used a Chinese version of the Eysenck Personality Questionnaire (EPQ, Cronbach’s alpha = 0.85) to assess the personality traits of patients with PD, which consists of 88 items and four subscales: psychoticism (P), extraversion (E), neuroticism (N), and lie (L) [59,60].

### 2.4. Imaging Data Acquisition and Preprocessing and ReHo Calculation

The imaging data were acquired using a 3.0 T Philips scanner with the following parameters: repetition time/echo time = 2000/30 ms, 33 slices, 64 × 64 matrices, 90° flip angle, 22 cm FOV, 4 mm slice thickness, no gap, and 240 volumes (480 s). During the scan, subjects were instructed to keep their eyes closed, remain motionless, and be awake without engaging in any specific thoughts. Earplugs and foam padding were used to restrict head movement and reduce noise. The imaging data were preprocessed using the Data Processing Assistant for Resting-state fMRI (DPARSF) toolbox 5.1 in MATLAB 2018a (http://www.mathworks.com, accessed on 20 August 2023) [61]. The first 10 time points of each subject were removed to allow for signal equilibrium and adaptation to the scanning environment. Then, slice timing and head motion correction were performed. Subjects with more than 2° angular motion in any direction or more than 2 mm displacement in the x-, y-, or z-axis were excluded from the analysis. The fMRI images were then normalized to the Montreal Neurological Institute (MNI) space with 3 mm × 3 mm × 3 mm resolution using the echo-planar imaging template [62]. Finally, the images were bandpass filtered (0.01–0.1 Hz) and linearly detrended [63].

The DPARSF toolbox was used to calculate ReHo. Kendall’s coefficient of concordance (KCC) was applied to synchronize the time series of each voxel with its 26 nearest voxels to obtain ReHo maps [64]. The KCC value was computed using the formula presented by Zang et al. [64]. To reduce the influence of individual variations, the KCC value of each voxel was divided by the mean KCC of the whole brain voxels for each subject. ReHo maps were then spatially smoothed using a 4 mm full width at half maximum Gaussian kernel to improve the signal-to-noise ratio [65].

### 2.5. Statistical Analysis

The demographic data and clinical symptoms were analyzed using SPSS 25.0 software, and statistical significance was set at *p* < 0.05. Differences between patients with PD and HCs were assessed using a chi-square test for gender distribution. Furthermore, for comparisons in age and educational level, we employed two-sample *t*-tests or Mann–Whitney U tests, which were selected based on the outcomes of the normality tests (Shapiro–Wilk tests) conducted on these variables [66]. Paired *t*-tests and Wilcoxon signed rank tests were employed to compare the clinical symptoms of patients with PD at baseline and after treatment, which were chosen based on the underlying nature of the variables. Additionally, we implemented Bonferroni’s correction to account for the multiple tests conducted and adjusted the *p*-values accordingly (*p* < 0.005 for the simultaneous conduct of 9 tests) [67]. Correlation between the ReHo of patients with PD at baseline and their clinical symptoms at baseline, as well as changes in ReHo and changes in clinical symptoms after treatment, were assessed using Spearman’s correlation for nonparametric data or Pearson correlation for normally distributed data. To account for multiple tests (9 conducted simultaneously), the *p*-value threshold was adjusted with the Bonferroni correction (*p* < 0.005).

The imaging data were analyzed using the DPARSF toolbox [61]. To identify clusters with abnormal ReHo in patients with PD at baseline compared to HCs, we performed two-sample *t*-tests on individual ReHo maps (Gaussian random field (GRF) correction, voxel *p*-value < 0.001, cluster *p*-value < 0.05, two-tailed), with gender, age, education level, and mean framewise displacement serving as covariates of no interest. To detect clusters with changed ReHo after treatment, we conducted paired *t*-tests on the pretreatment and posttreatment ReHo maps of patients with PD (GRF corrected, voxel *p*-value < 0.001, cluster *p*-value < 0.05, two-tailed).

### 2.6. SVR Analysis

This study utilized the LIBSVM toolbox 3.3 [68] to implement the support vector regression (SVR) algorithm to predict the treatment response of patients with PD to paroxetine based on their abnormal ReHo. The actual treatment response was measured by the reduction rates (RRs) of the HAMA and HAMD scores after treatment. To calculate the RR of the HAMA score, we used the following formula:(1)$$RR_{HAMA}=\frac{HAMA_{baseline}-HAMA_{4w}}{HAMA_{baseline}}$$

In Equation (1), RR_HAMA_ represents the reduction rate of the HAMA score after 4 weeks of treatment, HAMA_baseline_ represents the HAMA score at baseline, and HAMA_4w_ represents the HAMA score after 4 weeks of treatment. We used the same method to calculate the RR of the HAMD score. The SVR algorithm utilized 5-fold cross-validation with a radial basis function (RBF) kernel function, partitioning patients with PD into 5 subsets, with 80% of participants used for training and 20% for testing in each partition. Parameter optimization was conducted through a grid search method, systematically investigating a range of values for the “C” and “gamma” parameters in the SVR. The “C” parameter governs the balance between maximizing the margin between classes and minimizing classification errors, where smaller values encourage a larger margin but allow for some misclassification, while larger values prioritize accurate classification with a potentially narrower margin. Simultaneously, the “gamma” parameter shapes the decision boundary, determining the influence of individual training examples; lower “gamma” values produce a more general decision boundary, while higher “gamma” values lead to a more intricate and flexible boundary. These parameters were explored across a wide spectrum, spanning from 2^−10^ to 2^10^ [69]. This comprehensive search strategy aimed to identify the most optimal combination of “C” and “gamma” values for the SVR model within this range of potential parameter values. The predicted RRs of the HAMA and HAMD scores were aggregated across the 5 folds, and the performance of the model was evaluated using mean square error (MSE) and Pearson correlation between the predicted and observed treatment responses. The significance threshold of the Pearson correlation was corrected using the Bonferroni correction [*p* = 0.001(0.05/36)]. The study also conducted a permutation test to determine whether the correlation and MSE were significantly above chance [70]. This involved randomly reassigning the labels (observed RRs of the HAMA or HAMD scores) to the features (ReHo) 5000 times, creating reshuffled data that were then used to predict the RRs of the HAMA or HAMD scores in the same manner as the real data. This process generated 5000 MSEs and 5000 correlations between the predicted treatment response and the actual treatment response. To calculate the *p*-value for the real MSE or correlation, the number of permutations resulting in a smaller MSE or higher correlation than the real MSE or correlation was divided by the total number of permutations (5000). The SVR algorithms used in the study were previously described by Yan, Shan, Li, Liu and Guo [41].

## 3. Results

### 3.1. Demographic and Clinical Characteristics

We initially recruited 57 patients with PD and 56 HCs, but 3 patients and 2 HCs were excluded due to excessive head movement. As a result, the final analysis included 54 patients with PD and 54 HCs. The two groups were well-matched in terms of age (Mann–Whitney U test, *p* = 0.06), gender (chi-square test, *p* = 0.44), and educational level (Mann–Whitney U test, *p* = 0.76). Of the 54 patients with PD, 36 patients with PD completed the 4-week follow-up, while 18 patients with PD were unable to participate in the second resting-state fMRI scan due to the Coronavirus Disease 2019 pandemic. Detailed demographic and clinical data for all participants are presented in Table 1 and Appendix A.

### 3.2. Paroxetine Treatment Outcome

Appendix A and Appendix A present the clinical data of the patients with PD who completed the follow-up. The patients received paroxetine treatment for a mean duration of 34.81 ± 7.09 days. Results indicated that the patients experienced significant clinical improvement after treatment, as reflected in changes in scale scores. Specifically, notable changes were observed in the scores of HAMD (Wilcoxon signed rank test, *p* < 0.001), HAMA (Wilcoxon signed rank test, *p* < 0.001), SDSS (Wilcoxon signed rank test, *p* < 0.001, *df* = 35), and active coping (paired *t*-test, *p* = 0.017), along with various B-CATS subscales: DSST (paired *t*-test, *p* = 0.002), TMT-A (Wilcoxon signed rank test, *p* = 0.008), and TMT-B (Wilcoxon signed rank test, *p* = 0.004) after treatment. However, the significance of changes in active coping and TMT-A did not survive after the Bonferroni correction. Additionally, no significant alterations were identified in passive coping scores (paired *t*-test, *p* = 0.611) or CF scores (Wilcoxon signed rank test, *p* = 0.167) in PD patients.

### 3.3. ReHo Analysis Results

Compared to HCs, patients with PD exhibited lower ReHo in several brain regions, including the bilateral postcentral/precentral gyrus, bilateral fusiform gyrus/cerebellum VI, right calcarine/lingual gyrus, and left superior parietal lobule, and higher ReHo in the left superior/middle/inferior frontal gyrus (Table 2 and Figure 2). Significantly, after treatment, there was a notable increase in ReHo within the left fusiform gyrus in patients with PD when compared to their ReHo levels prior to treatment, as evident from the paired *t*-tests applied to the ReHo maps (Table 2 and Figure 3A). For further insight, ReHo values of the left fusiform gyrus were extracted from the baseline data of HCs, the baseline data of patients, the baseline data of patients who completed the follow-up, and the post-treatment data of patients. A comprehensive comparison was performed on these ReHo values using an ANOVA test (F = 10.16, *p* < 0.001, *df* = 3, r^2^ = 0.15) with Bonferroni-corrected post hoc *t*-tests (as depicted in Figure 3B). Notably, these results corroborated the findings obtained through the ReHo map comparisons.

### 3.4. Correlation between ReHo and Clinical Symptoms of Patients with PD

At baseline, there was a negative correlation between ReHo in the right fusiform gyrus/cerebellum VI of patients with PD and the TMT-A score (Pearson correlation, r = −0.286, *p* = 0.036, *df* =52, effect size: r^2^ = 0.08, Appendix A). However, this correlation did not survive the Bonferroni correction. No significant correlation was observed between baseline ReHo and other clinical symptoms in patients with PD.

The rate of change in ReHo in the left fusiform gyrus of patients with PD after treatment was not normally distributed (Shapiro–Wilk test: W = 0.884, *p* = 0.002). Therefore, Spearman’s correlation was used to examine the correlation between the rate of change in ReHo and the rates of change in symptom severity. The results indicated that after treatment, the rate of change in ReHo in the left fusiform gyrus of patients with PD was positively correlated with the RR of the HAMA score (Spearman’s correlation, rho = 0.385, *p* = 0.038, *df* = 34, effect size: r^2^ = 0.15, Figure 3C). However, this correlation did not remain significant after the Bonferroni correction.

### 3.5. SVR Analysis Results

SVR was used to test whether the abnormal ReHo of patients with PD before treatment could predict changes in their HAMA and HAMD scores after treatment. The predicted RR was significantly correlated with the actual RR of the HAMA score (Pearson r = 0.9469, corresponding *p* < 0.0001, *df* =34, effect size: r^2^ = 0.90, the *p*-value of permutation test < 0.0001, Figure 4A), with a low MSE (MSE = 0.0572, the *p*-value of permutation test < 0.0001, Figure 4A). Similarly, the predicted RR was significantly correlated with the actual RR of the HAMD score (Pearson r = 0.9927, corresponding *p* < 0.0001, *df* =34, effect size: r^2^ = 0.99, the *p*-value of permutation test < 0.0001, Figure 4B), with a low MSE (MSE = 0.0092, the *p*-value of permutation test < 0.0001, Figure 4B).

## 4. Discussion

The present study was the first longitudinal investigation to address several research questions, including (1) differences in ReHo between patients with PD and HCs; (2) changes in ReHo in patients with PD following pharmacotherapy; (3) relationships between ReHo and illness duration, depressive and anxiety symptoms, social and cognitive functions, coping strategies, and personality traits in patients with PD; (4) associations between changes in ReHo and changes in clinical symptoms of patients with PD; and (5) the potential for using baseline ReHo to predict early treatment response to paroxetine in patients with PD.

The study found that patients with PD had lower ReHo in the bilateral postcentral/precentral gyrus, bilateral fusiform gyrus and cerebellum VI, right calcarine/lingual gyrus, and left superior parietal lobule, as well as higher ReHo in the left superior/middle/inferior frontal gyrus, compared to HCs. After four weeks of treatment with paroxetine, the clinical symptoms of patients with PD improved significantly, and ReHo in the left fusiform gyrus increased substantially. The rate of increase in ReHo in the left fusiform gyrus was positively correlated with the reduction rate of the HAMA scores (Spearman’s correlation, rho = 0.385, *p* = 0.038). Moreover, ReHo in the right fusiform gyrus/cerebellum VI was negatively correlated with TMT-A scores in patients with PD (Pearson correlation, r = −0.286, *p* = 0.036). However, these two correlations did not survive the Bonferroni correction. The SVR analysis revealed small MSEs and significant positive correlations between the predicted and actual reduction rates of the HAMA or HAMD scores (*p* < 0.0001), indicating that abnormal baseline ReHo could be a predictor of early treatment response to paroxetine in patients with PD.

### 4.1. Abnormal ReHo at Baseline

In this study, we found that patients with PD had decreased ReHo values in the bilateral postcentral/precentral gyrus compared to HCs, consistent with previous research indicating aberrant sensorimotor network activity in PD patients [71,72]. Prior study has also suggested that treatment with escitalopram can improve reduced activation in the sensorimotor network of patients with PD [73]. The postcentral gyrus, which is responsible for receiving, integrating, and interpreting somatic sensations [74], showed decreased ReHo values in our study, indicating abnormal processing of somatosensory stimuli and potential misinterpretation of somatic sensations. This could, in turn, activate the fear network [13], and may explain the somatosensory abnormalities observed in PD patients, such as heat or cold sensation and numbness. The precentral gyrus is involved in motor and somatosensory functions and the regulation of anticipatory threat and anticipatory anxiety [75]. Patients with PD showed hypoactivity in the precentral gyrus during both rest and anticipatory anxiety, although more voxels were deactivated during anticipatory anxiety, as reported by Boshuisen, et al. [76]. Additionally, Lai and Wu [77] found increased GMV in the left superior frontal gyrus and residual GMV deficits in the precentral gyrus of remitted PD patients after 6 weeks of escitalopram treatment, suggesting that alterations in the precentral gyrus could be a trait marker in PD [78]. To summarize, the presence of an abnormal sensorimotor network in PD patients may contribute to their susceptibility to panic attacks and help to explain their somatosensory abnormalities and anticipatory anxiety.

Our research found that compared to HCs, patients with PD had lower ReHo in the bilateral fusiform gyrus and right calcarine/lingual gyrus, which are visually associated cortices. Previous studies have also reported abnormalities in the fusiform gyrus of PD patients, including weakened causal connectivity between the stria terminalis and the right fusiform gyrus [79] and cortical thinning in the right fusiform gyrus [80]. The fusiform gyrus is a crucial structure for high-level vision tasks, such as face perception [81], object recognition [82], and reading [83]. Furthermore, the fusiform gyrus is a significant site for associative fear learning and remote fear memory storage [84], and the amygdala modulates its activation in response to threat stimuli [85]. The decreased ReHo in the bilateral fusiform gyrus of PD patients may underlie the dysfunction of visual processing to threat stimuli and impaired associative fear learning, which are linked with panic responses while patients with PD experience environmental stressors [86]. In addition, our study found lower ReHo in the right calcarine/lingual gyrus of patients with PD, which aligns with previous research [49,87]. The lingual gyrus and the amygdala collaborate in spatial attention and fear processing [88]. Moreover, The lingual gyrus is involved in vigilance [89], cardiovascular functions [90], and anticipatory anxiety [91], while the calcarine gyrus, the terminal of neural impulses originating in the retina and the site of visual signal transition, mediates visuo-sensory function [92]. The lower ReHo in the visual cortex including the bilateral fusiform gyrus and right calcarine/lingual gyrus, may be implicated in anticipatory anxiety, visual processing dysfunction regarding threat stimuli, impaired associative fear learning, and dysregulation of autonomic function in PD patients [93,94].

In this study, at baseline, patients with PD demonstrated decreased ReHo in the left superior parietal lobule and increased ReHo in the left superior/middle/inferior frontal gyrus compared to HCs. Previous studies reported other abnormalities in the fronto-parietal areas of patients with PD relative to HCs, including reduced cortical gyrification [24], cortical thickness [48], resting-state FC [95], degree centrality [96], and nodal efficiency [97]. Additionally, task-based fMRI studies have identified aberrant activity in the fronto-parietal pathways of patients with PD during interoceptive [98], panic-motivated [50,99], or alerting tasks [30]. Jin, Zhang, Cui, Li, Li, Hu, Wang and Li [98] found increased activity in the bilateral superior parietal lobule of patients with PD relative to HCs during the heartbeat perception task. Furthermore, in PD patients, left superior parietal lobule BOLD activity during the heartbeat perception task was positively correlated with heartbeat perception scores, suggesting the superior parietal lobule’s involvement in perceptive processing, particularly interoceptive awareness relevant to PD [98]. Lopes, Faria, Dias, Mallmann, Mendes, Horato, de-Melo-Neto, Veras, Magalhães, Malaspina and Nardi [50] and Kircher, et al. [100] reported increased neural activation in the frontal gyrus of PD patients relative to HCs during panic-motivated tasks, indicating the frontal gyrus’s role in fear acquisition and conditioning. Neufang, Geiger, Homola, Mahr, Schiele, Gehrmann, Schmidt, Gajewska, Nowak and Meisenzahl-Lechner [30] reported reduced activation in the fronto-parietal pathways, including the middle frontal gyrus and superior parietal lobule, in patients with PD during the attention network task, which involves detecting targets and cues quickly. They also found that FC within the fronto-parietal area was negatively correlated with Anxiety Sensitivity Index scores [30]. The findings of Neufang, Geiger, Homola, Mahr, Schiele, Gehrmann, Schmidt, Gajewska, Nowak and Meisenzahl-Lechner [30] indicate that the fronto-parietal areas are key structures for alerting to external and internal stimuli. Therefore, aberrant neural activity in the frontal–parietal regions of individuals with PD can lead to impaired perceptive processing, fear conditioning, and increased alertness to both external and internal stimuli.

In this study, it was found that patients with PD had reduced ReHo in the bilateral cerebellum VI compared to HCs. The cerebellum is known to be involved in a variety of processes, including sensorimotor, cognitive, and affective functions [101], with cerebellum VI specifically related to cognitive function [102]. Additionally, a negative correlation was observed between TMT-A scores and ReHo values in the right fusiform gyrus and cerebellum VI, indicating that patients with PD with lower ReHo in these regions may have poorer cognitive performance. Therefore, the decreased ReHo in the bilateral cerebellum VI observed in patients with PD may explain their impaired cognitive functions [47].

To summarize, the present study revealed abnormal ReHo in several brain regions in patients with PD compared to HCs, including the sensorimotor network, visual cortex, frontal–parietal areas, and cerebellum. According to the hypothesis of Lai [19] on the advanced fear network for PD, the sensory regions, including temporal, parietal, and occipital lobes, send sensory information to the thalamus, which filters the information before the insula integrates it for cognitive processing by the frontal gyrus and generating a primitive response by the limbic system. However, we did not find abnormal ReHo in the limbic system in the present study. This might be attributed to the aberrant activity of the limbic system as a state characteristic rather than a trait characteristic of PD [18]. Consistent with the hypothesis of Lai [19] our findings suggest that abnormal sensory processing, including somatosensory, interoceptive awareness, and visual–spatial function, may activate the frontal–parietal attentional or alerting system and the cognitive-affective response of the cerebellum, ultimately influencing the cognitive processing of the frontal gyrus and generating a primitive response in the limbic system. Therefore, our results support the notion that brain networks involved in somatosensory processing, visuospatial function, interoceptive awareness, and emotional regulation play a crucial role in the pathophysiology of PD.

### 4.2. Changes in ReHo after Treatment

After four weeks of paroxetine treatment, a significant increase in ReHo was observed in the left fusiform gyrus, which had shown reduced ReHo before treatment. Moreover, the rate of increase in ReHo was positively correlated with the reduction rate of HAMA scores (Spearman’s correlation, rho = 0.385, *p* = 0.038). However, this correlation did not survive the Bonferroni correction. Consistent with these findings, normalization of brain function after treatment was observed in patients with PD who received SSRIs [26] or psychodynamic treatment [25,30]. For instance, Lai and Wu [26] reported that decreased ReHo in the right Heschl gyrus increased remarkably in remitted patients with PD who received a 6-week treatment with escitalopram. Additionally, Beutel, Stark, Pan, Silbersweig and Dietrich [25] observed that abnormal fronto-limbic activation patterns in patients with PD, detected during emotional linguistic go/no-go tasks, were normalized after 4-week psychodynamic inpatient treatment. Similarly, after 6 weeks of CBT, decreased middle frontal and parietal activation of patients with PD during attention network tasks increased remarkably [30]. As previously mentioned, decreased ReHo observed in the fusiform gyrus of patients with PD at baseline might be at the root of compromised visual processing of threat stimuli and impaired associative fear learning. These cognitive deficits have been associated with heightened panic responses in PD patients exposed to environmental stressors [86]. Consequently, the restoration of ReHo in the fusiform gyrus would signify the reinstatement of normal visual processing for threat stimuli and improved associative fear learning. This issue, in turn, could lead to a reduction in panic responses when PD patients confront environmental stressors. By normalizing these cognitive functions and mitigating panic responses, the severity of anxiety symptoms is anticipated to decrease. As a result, there is a positive correlation between the rate of increase in ReHo and the rate of reduction in HAMA scores. A comparable pattern emerged in individuals with schizophrenia. Yang et al. found that prior to treatment, patients diagnosed with schizophrenia displayed an extended range of global gradient scores, a method used to identify hierarchical alterations, in contrast to HCs [103], suggesting a form of functional segregation within subcortical systems. The heightened gradient observed in the limbic system, coupled with the reduced gradient within the thalamic and striatal system, contributed to the initial abnormalities, ultimately resulting in the disruption of subcortical functional integration [103]. Following the treatment intervention, these disruptions underwent a process of normalization. Notably, the longitudinal shifts in gradient scores within the limbic system exhibited a significant association with the amelioration of symptoms. This association was particularly pronounced in relation to the enhancement of the disorganization and excitement symptom domain [103]. The findings of the present study, which show the normalization of ReHo in the left fusiform gyrus of PD patients after 4 weeks of paroxetine treatment and a correlation between this change and the improvement of anxiety severity, suggest the important role of the left fusiform gyrus in the pharmacological mechanism underlying paroxetine treatment of PD. Furthermore, these functional brain changes in patients with PD who received pharmacotherapy were attributed to the treatment effects.

### 4.3. Applying SVR to Predict the Treatment Response from Abnormal ReHo

Our SVR analysis demonstrated the low MSE and the significant positive correlation between the predicted RR and actual RR of both HAMA and HAMD scores, indicating that abnormal ReHo at baseline could be applied to predict the early treatment response of patients with PD to paroxetine. Previous studies reported other machine learning algorithms could be used to predict the treatment response of patients with PD to pharmacotherapy or CBT from brain structural or functional features [32,104]. For example, Lasso regression could be applied to predict the treatment response of patients with PD to pharmacotherapy from fractional anisotropy values of white matter [104], and Gaussian process classifiers could be employed to predict the treatment response of patients with PD to CBT from the BOLD signal measured during a fear-conditioning task [32]. These multivariate analyses, including SVR, take into account the temporal and spatial patterns of brain networks relative to conventional univariate methods and could predict the treatment response at the individual level. The findings of our SVR analysis have important clinical implications for precision medicine, particularly in identifying which patients with PD will derive the most benefit from paroxetine treatment. The use of prognostic biomarkers could help to reduce treatment duration and improve response rates, leading to better outcomes for patients.

### 4.4. Limitations

The present study has several limitations that should be acknowledged. First, the notable attrition rate throughout the follow-up phase and the occurrence of depressive symptoms among certain PD patients might have introduced bias to the findings [105]. We reached out to individuals who were unable to engage in the follow-up process and ascertained that a significant portion of them withdrew due to the disruptions brought about by the coronavirus disease 2019 pandemic. Second, due to HCs being scanned only at baseline, a repeated-measure comparison for changes in ReHo of HCs was not possible. Therefore, the changes in ReHo observed in patients with PD after treatment may not all be attributed to the treatment effects of paroxetine. Nevertheless, previous studies have demonstrated that ReHo is a relatively stable resting-state fMRI indicator, and intraindividual fluctuations are limited in HCs [106]. Third, considering the dynamic and state-dependent nature of fMRI, incorporating structural MRI indicators within a longitudinal study could enhance the validation and extension of our findings, particularly in observing the evolving brain changes following treatment. Last, while the SVR was successful in predicting treatment response based on abnormal ReHo, our data were collected from a single center and a relatively homogeneous population. It remains unclear whether similar results could be achieved using multicenter data from subjects with a diverse population and/or comorbidities.

## 5. Conclusions

In summary, this study identified abnormal spontaneous neural activities in the sensorimotor network, visual cortex, frontal–parietal pathway, and cerebellum of patients with PD, which validated the advanced fear network model proposed by Lai [19]. These abnormal neural activities at baseline could serve as candidate predictors for treatment response to paroxetine in patients with PD. Furthermore, our results showed that abnormal spontaneous neural activities in patients with PD could be partially normalized with pharmacotherapy, suggesting that the functional brain changes observed in the early treatment stage were likely attributed to the treatment effects of paroxetine. Future long-term studies could investigate if pharmacotherapy or psychotherapy normalizes all observed abnormal brain activity in PD patients, predicting long-term effects and sustaining early treatment normalization. Gaining insight into the trajectories of brain activity normalization following treatment holds the potential to provide vital insights for managing PD.

## Figures and Tables

**Figure 1 biomedicines-11-02420-f001:**
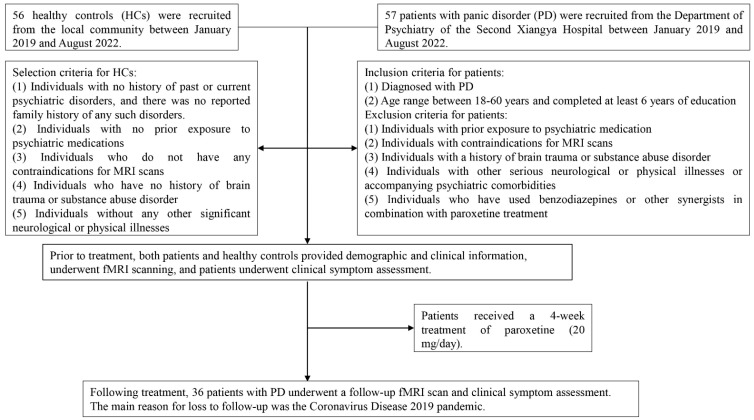
Study flow chart.

**Figure 2 biomedicines-11-02420-f002:**
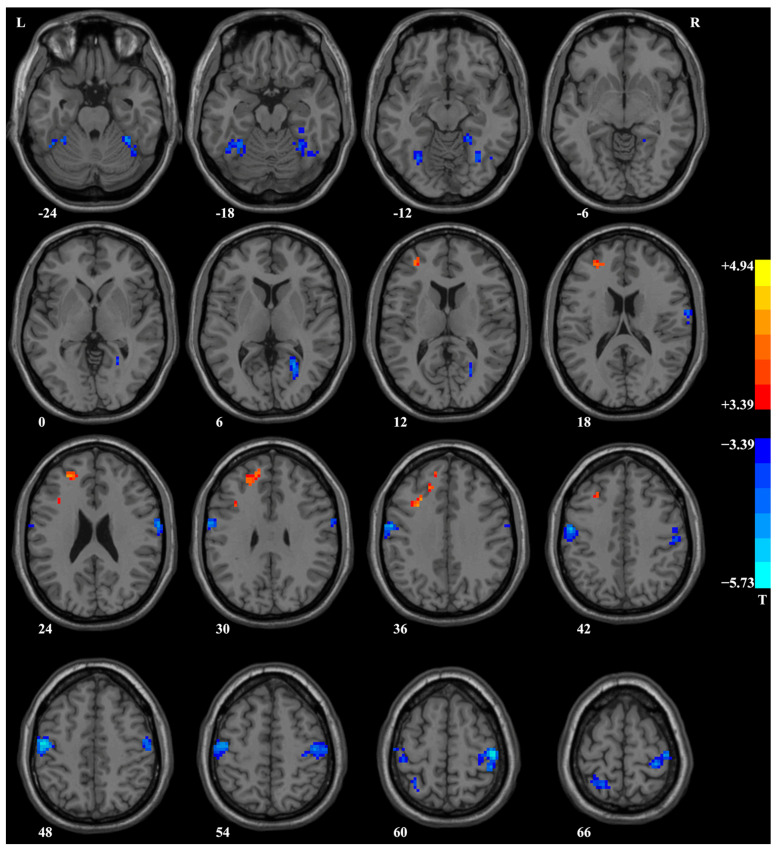
Brain regions with a significant difference in ReHo between patients with PD and HCs. Compared to HCs, patients with PD showed lower ReHo in the bilateral postcentral/precentral gyrus, bilateral fusiform gyrus/cerebellum VI, right calcarine/lingual gyrus, and left superior parietal lobule and higher ReHo in the left superior/middle/inferior frontal gyrus. ReHo, regional homogeneity; PD, panic disorder; HCs, healthy controls.

**Figure 3 biomedicines-11-02420-f003:**
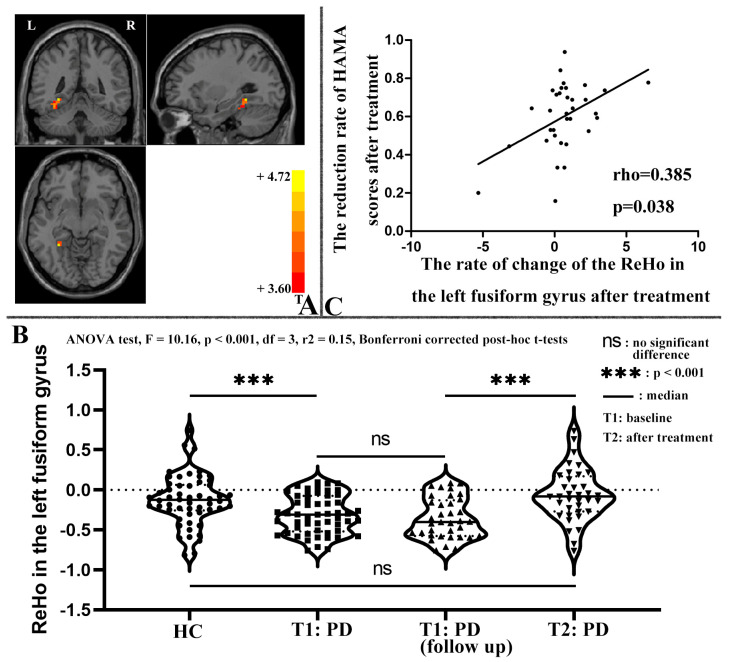
Part (**A**): Brain region with a significant change in ReHo after treatment. Patients with PD showed significantly increased ReHo in the left fusiform gyrus after four weeks of paroxetine treatment compared to baseline data. Part (**B**): A thorough comparison was conducted on the ReHo values of the left fusiform gyrus, which were derived from the baseline data of healthy controls, baseline data of patients, baseline data of patients who successfully completed the follow-up, and post-treatment data of patients. Each circle, square, and triangle represents a participant. Part (**C**): The rate of change in the ReHo in the left fusiform gyrus after treatment was positively correlated with the reduction rate of HAMA scores (Spearman’s correlation, rho = 0.385, *p* = 0.038). The least square fitted line is shown in black. ReHo, regional homogeneity; HC, healthy control; PD, panic disorder; HAMA, Hamilton Anxiety Rating Scale.

**Figure 4 biomedicines-11-02420-f004:**
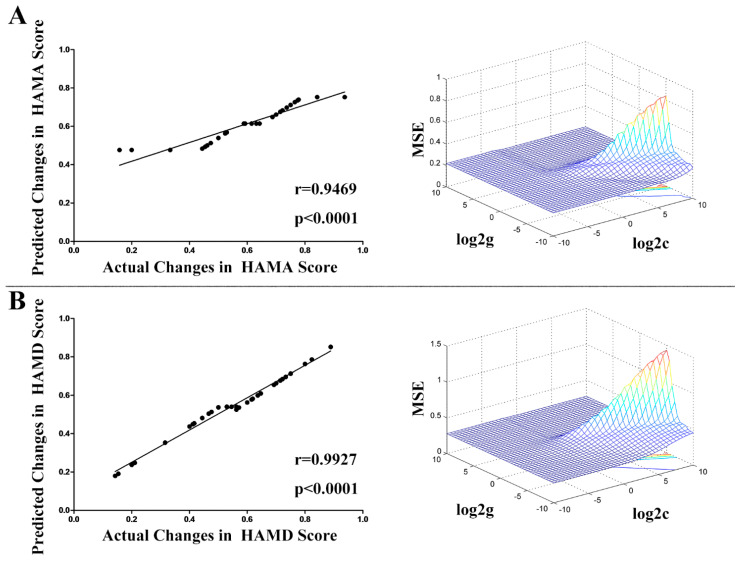
Evaluation of the performance of the predictive model by calculating the MSE and Pearson correlation between the predicted treatment response (predicted RRs of the HAMA and HAMD scores) and the actual treatment response (actual RRs of the HAMA and HAMD scores). Part (**A**): The predicted RR of the HAMA score was significantly correlated with the actual RR of the HAMA score (Pearson r = 0.9469, *p* < 0.0001), and the MSE was low (MSE = 0.0572). Part (**B**): The predicted RR of the HAMD score was significantly correlated with the actual RR of the HAMD score (Pearson r = 0.9927, *p* < 0.0001), and the MSE was low (MSE = 0.0092). SVR parameters used were “c” for the “C” parameter and “g” was the “gamma” parameter of SVR. The varying colors of the lines indicate distinct MSE values. Warmer colors signify higher MSE, while cooler colors denote lower MSE. MSE, mean square error; RR, reduction rate; HAMA, Hamilton Anxiety Rating Scale; HAMD, Hamilton Depression Rating Scale; SVR, support vector regression.

**Table 1 biomedicines-11-02420-t001:** Characteristics of participants.

Variables	Patients (Mean ± SD, n = 54)	Controls (Mean ± SD, n = 54)	*U*/*χ^2^*	*p*-Value	*df*	Effect Size: *r*/Cramer’s V
Age (years)	34.78 ± 9.67	32.28 ± 10.56	1151.50	0.06 ^a^	106	−0.18
Sex (male/female)	25/29	21/33	0.61	0.44 ^b^	1	0.08
Years of education (years)	13.15 ± 3.45	13.43 ± 3.22	1410.00	0.76 ^a^	106	−0.03
Illness duration (months)	14.22 ± 21.88					
HAMD	14.07 ± 4.01					
HAMA	17.59 ± 4.37					
SDSS	3.28 ± 3.31					
CSQ						
Active coping	20.04 ± 6.24					
Passive coping	11.35 ± 4.38					
B-CATS						
Digit symbol substitution	52.57 ± 17.05					
Trail making test part A	38.31 ± 15.87					
Trail making test part B	75.59 ± 43.38					
Category fluency	17.89 ± 6.73					
EPQ						
E	50.74 ± 9.83					
P	65.28 ± 15.49					
N	45.09 ± 9.39					
L	39.91 ± 12.83					

^a^ The *p*-values were obtained by Mann–Whitney U tests. ^b^ The *p*-value for sex distribution was obtained by a chi-square test. SD = standard deviation; HAMD = Hamilton Depression Rating Scale; HAMA = Hamilton Anxiety Rating Scale; SDSS = Social Disability Screening Schedule; CSQ = Simplified Coping Style Questionnaire; B-CATS = Brief Cognitive Assessment Tool for Schizophrenia; EPQ = Eysenck Personality Questionnaire; E = Extraversion; P = Psychoticism; N = Neuroticism; L = Lie.

**Table 2 biomedicines-11-02420-t002:** Regions with abnormal ReHo in patients with PD at baseline and alterations of ReHo after treatment.

Cluster Location	Peak (MNI)			Number of Voxels	*T* Value	*p*	*df*	Cohen’s d
	x	y	z					
Patients with PD at baseline versus controls								
Right postcentral/precentral gyrus	51	−18	42	253	−3.39	<0.001	106	−0.65
Left postcentral/precentral gyrus	−57	−6	48	220	−3.39	<0.001	106	−0.65
Right fusiform gyrus/cerebellum VI	18	−45	−12	121	−3.39	<0.001	106	−0.65
Left superior/middle frontal gyrus	−27	45	21	88	4.43	<0.001	106	0.85
Left fusiform gyrus/cerebellum VI	−30	−48	−24	85	−3.40	<0.001	106	−0.65
Right calcarine/lingual gyrus	27	−57	9	58	−3.42	<0.001	106	−0.66
Right postcentral gyrus	63	−6	33	53	−3.40	<0.001	106	−0.65
Left superior parietal lobule	−15	−57	66	44	−3.41	<0.001	106	−0.66
Left middle/inferior frontal gyrus (triangular part)	−30	15	33	38	4.94	<0.001	106	0.95
Patients with PD after 4-week treatment versus at baseline								
Left fusiform gyrus	−30	−45	−9	41	4.57	<0.001	35	1.08

PD = panic disorder; ReHo = regional homogeneity; MNI = Montreal Neurological Institute.

## Data Availability

The data that support the findings of this study are available from the corresponding author, Wenbin Guo, upon reasonable request.

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
