# Peer review of "Breaking the Fear Barrier: Aberrant Activity of Fear Networks as a Prognostic Biomarker in Patients with Panic Disorder Normalized by Pharmacotherapy"

_biomedicines, 2023, doi:10.3390/biomedicines11092420_

Round 1
Reviewer 1 Report
Appraisal: The introduction is quite confusing. The etiopathogenesis of panic disorder should be explained I a better way. The authors, indeed, stated that PD is a complex disorder with a complex etiopathogenesis. However, they cited only one ref. This is crucial to explain the role of brain regions such as Amygdala etc. Moreover, the conceptual order of the introduction is not clear. The authors introduced the resting state fMRI after the explicative models (only one and the reason was not really explicit) and then they introduced the principal therapy for the PD. This need to be rewritten or organized in a better way. The methods and the flowchart are explicative and allow to replicate the study.” HCs had no past or current 121 Axis I conditions and no family history of psychosis.” Why the authors did not take into account family history of PD? “fMRI scans on them using a 3.0 T MRI scanner” The authors need to write more details about the scanner in the main text.” Following the baseline scans, patients with PD received 4 weeks of treatment with paroxetine (20 mg/day), and underwent a second scan 135 after the treatment period.” The patients received 20 mg /day of Paroxetine. However, It is not clear if the patients were in pharmacological washout. Usually, PD is treated with 20 mg /day Paroxetine and maybe most of the patients experimented this therapy in the past. The authors only used PD with 20 mg /day of Paroxetine treatment but not with placebo. Why? Please explain.
Moreover, it should be interesting if the patients, or some of them, also took alprazolam. Measures: It is not clear if the patients were assessed before and after the treatment. Please clarify. According to me, the acquisition parameters cannot be included in the supplementary materials, because they are important for the replication of the study. Moreover, the band pass filter (0.01-0.1 Hz- I agree) needs a ref. Neuroimaging community also reads the present journal and more details are needed.
Statistical analyses should be more complex. Indeed, t test is useful for small samples, but the authors collected more than 100 total participants. I advise a mixed model ANOVA/ MANOVA . Furthermore, the chi sqr results should be corrected too.
“Parameter optimization was conducted using a grid search method to determine the optimal values for the "C" 189 and "gamma" parameters of SVR” I know what you mean, but this statement about C and gamma should be written in a better way. “Notably, after treatment, the ReHo in the left fusiform gyrus of patients with PD increased significantly compared to pretreatment ReHo (Table 2 and Figure 3A)” Authors wrote that the PD after treatment were compared (ReHo) with the pre-treatment. According to me, it is not correct and the comparison should be : (PD( Pre) >HC)> ( PD (post) < HC). Please check.
Reviewer 2 Report
Aberrant activity of fear networks as a potential prognostic biomarker in patients with panic disorder normalized by pharmacotherapy (biomedicines-2535296)
(Review)
Main message of the article
Yan and colleagues used functional magnetic resonance imaging to investigate resting-state brain activity in patients with panic disorder before and after pharmacotherapy. Different patterns of brain activity emerged when comparing patients with panic disorder to healthy controls. After 4 weeks of paroxetine treatment, brain activity increased in the left fusiform gyrus and support vector regression predicted treatment response.
General Judgment Comments
The article is clearly written and well represented by the title and the keywords selected by the authors. The Abstract does not provide much theoretical background to help the reader understand the context in which the research develops. As regards the Methods, more details need to be provided about the instruments’ reliability, the choice of the sample size (especially considering the data loss of 18 participants), and the statistical approach (e.g., the use of Bonferroni’s correction, assumptions tests for parametric statistics). Furthermore, more details on the fMRI data acquisition, pre-processing, and ReHo computation need to be provided. Results are not reported following the standard format. Figures can be improved by increasing the font size and by removing colors where they are not needed to facilitate the reading for people with visual deficits. Ultimately, the correlation between anxiety symptoms and ReHo is quite misleading throughout the manuscript and the authors need to specify that this result is non-significant based on Bonferroni’s correction.
I recommend for Major Revision.
Major Issues
-
- Abstract: no theoretical background is provided.
-
- Abstract: how is the treatment administered? How often?
-
- Abstract: Lines 23-25 are misleading as this result did not survive Bonferroni correction. Please specify that this result is non-significant in the Abstract. This applies also to Lines 405-407 of the discussion.
-
- Introduction: line 70 needs to be justified with references.
-
- Participants: how did the authors decide how many participants to include in their study?
-
- Lines 138-154: please justify the choice of the instruments and provide the Cronbach’s alphas.
-
- Line 155: the authors should clarify the details of data acquisition, preprocessing and ReHo calculation in the main text because these are crucial art to understand and replicate the study.
-
- Statistical analysis: “Differences between patients with PD 160 and HCs in gender distribution, age, and educational
level were evaluated using chi-square tests or two-sample t-tests”. Before running the t-tests, were the assumptions checked? If so, authors need to clarify and report the results for the assumptions checks.
-
- Lines 162-164: were the assumptions of the t-test checked? Furthermore, the authors should use the Bonferroni’s correction to adjust the p-value for the number of tests that were conducted to assess differences in clinical symptoms
at baseline and after treatment.
-
- Line 167: specify the p-value after the Bonferroni’s correction.
-
- The loss of 18 participants in the post-treatment phase is quite relevant.
-
- Results: please report the results following the standard format: the p-value is not enough, and it needs to be presented together with the statistic of interest, the degrees of freedom, the confidence intervals, and the effect size.
Minor Issues
-
- Line 38: DSM-5 is repeated.
-
- Figure 1: the font is too small, and the quality of the figure should be improved.
-
- Lines 119-121: the references to the instruments are missing.
-
- Lines 138-154: references to the instruments are missing.
-
- Figure 3: colors should be removed from the scatterplot, especially if using green and red (Allred, S., Schreiner, W. & Smithies, O. Still too many red–green figures. Nature 510, 340 (2014)). I suggest using black for clarity.
- Figure 4: the font is too small and the choice of colors in the scatterplot and in the presentation of r and p-values is not clear. Please use black if possible.
Final comments
I recommend for Major Revision.
Reviewer 3 Report
Please read the attachment. Thank you.

Round 2
Reviewer 1 Report
Authors addressed all the issues that I raised.
Please, correct the typos in the manuscript.
Author Response
Thank you. We have addressed the typos in the manuscript. Please kindly refer to the revised version of the manuscript for updates.
Reviewer 2 Report
the article is now acceptable
the article is now acceptable
Author Response
Thank you.